# Compact Encoding of Words for Efficient Character-level Convolutional Neural Network Text Classification

## Abstract

This paper puts forward a new text to tensor representation that relies on information compression techniques to assign shorter codes to the most frequently used characters. This representation is language-independent with no need of pretraining and produces an encoding with no information loss. It provides an adequate description of the morphology of text, as it is able to represent prefixes, declensions, and inflections with similar vectors and are able to represent even unseen words on the training dataset. Similarly, as it is compact yet sparse, is ideal for speed up training times using tensor processing libraries. As part of this paper, we show that this technique is especially effective when coupled with convolutional neural networks (CNNs) for text classification at character-level. We apply two variants of CNN coupled with it. Experimental results show that it drastically reduces the number of parameters to be optimized, resulting in competitive classification accuracy values in only a fraction of the time spent by one-hot encoding representations, thus enabling training in commodity hardware.

## 1 Introduction

Document classification is one of the principal tasks addressed in the context of natural language processing (Sebastiani, 2002). It implies associating a document —or any text fragment, for that matter— with a category or label relying on their content. The increasing availability of texts in digital form, especially through the Internet, has called for the development of statistical and artificial intelligence tools for automating this process. Spam detectors, sentiment analysis, news archiving, among many others, demand high-quality text classifiers.

There is a broad range of approaches to document classification (see (Sebastiani, 2002; Aggarwal & Zhai, 2012; Hotho et al., 2005; Kosala & Blockeel, 2000)). An important portion of them relies on a representation that handles words as the atomic element of text. Consequently, those methods carry out their analysis through statistics of words occurrence (Zhang et al., 2015). However, the variability of words and structures belonging to a language hinders the viability of this method. That is why, these models have a superior performance in specific domains and applications, where the vocabulary is or can be restricted to a relatively small number of words, possibly chosen by a specialist. Furthermore, such modeling becomes specific to a language, causing the replication process in another language to be carried out from scratch (Zhang et al., 2015).

In recent years, we have experienced a revolution in the machine learning with the advent of deep learning methods (Goodfellow et al., 2016). The development of convolutional neural networks (CNNs) (LeCun et al., 1998) coupled with the popularization of parallel computing libraries (e. g. Theano (Bergstra et al., 2010), Tensorflow (Abadi et al., 2015), Keras (Chollet et al., 2015), etc.) that simplify general-purpose computing on graphics processing units (GPGPU) (Mittal & Vetter, 2015) has been successful in tackling image classification problem (Krizhevsky et al., 2012) quickly becoming the state of the art of the field.

As it could be expected, the success of deep learning and CNNs in the image classification domain has prompted the interest to extend the deep learning principles to the document classification domain. Some existing methods have been updated but the clear majority are still based on the to-

kenization of words and the inference of their statistics. Bag of Words (BoW) (Johnson & Zhang, 2014) and Word2vec (Mikolov et al., 2013) are some of the most popular strategies.

It can be argued that the replication of image classification success in the documents domain faces as main challenge the difficulty of representing text as numerical tensors.

To address this issue, Zhang et al. (2015) suggested a groundbreaking approach that considers the characters as the atomic elements of a text. In particular, they represented the text as a sequence of one-hot encoded characters. This encoding provides a robust, language-independent representation of texts as matrices, that are then used as inputs of different CNNs. Their experimental results showed that this approach was able to attain and, in some cases, improve the state of the art results in complex text classification problems. More recently, Xiao & Cho (2016) improved those results by combining CNNs with Long Short-Term Memories (LSTMs) (Hochreiter & Schmidhuber, 1997). In spite of that, the impact of this idea is hampered by the large computational demands of the approach, since its training can take days per epoch in relatively complex problems.

Character-level representations have the potential of being more robust than word-level ones. On the other hand, they are computationally more expensive because detecting syntactic and semantic relationships at the character-level is more expensive (Blunsom et al., 2017). One possible solution could be a word representation that incorporates the character-level information.

In this paper, we propose an efficient character-level encoding of word to represent texts derived from the Tagged Huffman (Silva de Moura et al., 2000) information compression technique. This encoding takes into account the character appearance frequency in the texts in order to assign shorter codes to the most frequently used ones. This novel text encoding makes the idea put forward by Zhang et al. (2015) more computationally accessible by reducing its training requirements in terms of time and memory.

The proposed encoding makes possible to represent larger portions of texts in a less sparse form, without any loss of information, while preserving the ability to encode any word, even those not present in the training dataset ones. In order to study the impact of this encoding, we coupled it with two CNN architectures. The experimental studies performed showed that we managed to achieve a performance similar or in some cases better than the state of the art at a fraction of the training time even if we employed a simpler hardware setup.

Our main contribution is to show that this novel character-level text encoding produces a reduced input matrix, leading to a substantial reduction in training times while producing comparable or better results in terms of accuracy than the original approach by Zhang et al. (2015). This opens the door to more complex applications, the use of devices with lower computational power and the exploration of other approaches that can be coupled with input representation.

The rest of the paper is structured as follows. In the next section, we deal with the theoretical foundations and motivation that are required for the ensuing discussions. There we also analyze the alternatives to character-level text compression that were taken into account for producing our proposal. After that, in Section 3, we describe the encoding procedure and the neural network architectures that will take part of the experiments. Subsequently, in Section 4, we replicate the experiments of Zhang et al. (2015) in order to contrast our proposal with theirs under comparable conditions. Finally, in Section 5, we provide some final remarks, conclusive comments and outline our future work directions.

## 2 PRELIMINARIES

The success in using Convolutional Neural Networks (CNNs) (LeCun et al., 1998) in image classification (Krizhevsky et al., 2012) flourish with the development of many libraries (Chetlur et al., 2014; Chollet et al., 2015; Bergstra et al., 2010), techniques and hardware. The effort to use CNNs for text classification tasks is justified by the possibility of appropriating these tools for obtaining better results and more robust algorithms, facilitating the use of the same approach for several applications.

There are two usual approaches to use CNNs for handling textual information: the (i) bag of words (BoW) (Johnson & Zhang, 2014) and (ii) Word2vec (Mikolov et al., 2013) approaches.

In the case of BoW and some of its variants, for representing each word in a vocabulary of size $N$, a digit 1 is placed in the correspondent position of that word in a $1 \times N$ vector, all others positions remaining with digit 0. Since natural languages usually have a large vocabulary, a limited subset of the vocabulary must be used in order to make viable to perform the necessary computations in terms of memory requirements. The chosen subset of the vocabulary must be representative of the texts. Therefore, in practical problems, a great deal of attention is devoted to this matter. In particular, it is common to involve an application domain specialist or use some kind of word frequency statistic or relevance metric, where the most frequent and rare words are excluded.

In the word2vec approach, each word is projected via a metric embedding of fixed size, representing its co-occurrence in a large corpus of text in the same language of the texts of interest. It is possible to use pretrained vectors or readjust the representation with new words. The main problem with both strategies is that it does not allows to represent words that are not in the training dataset. Typos, spelling errors, mixed up letters and text written in languages with a complex structure (declensions, conjugations, etc.) are completely ignored.

Establishing the character as the basic unit of text formation provides a better opportunity to be robust to typos, acceptance of neologisms, and other textual forms as equations and chemical formulas, abbreviations and idiosyncrasies of written language on the internet, such as emoticons, slang and dialects of the online world, etc. Assuming the word as a base item, much of this ability is lost, especially when models assume that text producers use a formal language.

Zhang et al. (2015) put forward an important innovation in this regard. They represent text as a sequence of characters, not words. Consequently, they are capable of reducing the vocabulary of symbols to the size of the alphabet (69 in the case of the paper) and thus allowing the use of one-hot encoding (Harris & Harris, 2012). In this paper, they represented a text as a matrix of size $1014 \times 69$ where each row corresponds to a position in the text sequence and columns with the presence or not of a given character. Therefore, a row with a 1 in a given column indicates that the presence of the corresponding character in that point of the text. With this representation on hand, they applied a CNN and obtained results competitive with other techniques, and in some cases improving the state of the art. However, the main drawback is a large computational requirement, that in some cases called for days per training epoch.

The results obtained by them suggest that language, and therefore, text, can be treated as a sequence of signals, like any other (Zhang et al., 2015). However, the training times and the dimension of the matrices to be computed are still obstacles to the most effective use of the method. That is why a better encoding of text could be a right path towards a substantial improvement if this issue.

## 3 COMPRESSED ENCODING FOR CNN-BASED TEXT CLASSIFICATION

Searching for a way to reduce these training time while retaining the flexibility and power of character-level convolutional to classify text, we found a way to better encode texts. Our approach, with competitive accuracy, achieve a significant reduction in time execution from hours to minutes and from days to hours, by epoch.

To achieve this performance in execution, our approach consisted of two elements:

- *Obtaining a shorter representation*: At first, we though of using some form of encoding each character, and use the same approach of Zhang et al. (2015) but we realize that using a variable length encoding for each word could be more efficient. To do this we need a way to encode each char, generating distinct concatenated code for representing each word and that words with the same prefix were near each other, especially to respond well to declensions.

- *Obtaining a sparse representation*: To take full advantage of libraries of tensor multiplications like NVidia CuDNN (Chetlur et al., 2014), we need a representation that was sparse, so our code should be composed of many 0s and a few 1s.

### 3.1 Compressed Representations

Although the Huffman encoding (Huffman, 1952) yields shortest possible, it does not generate unique representations once we concatenate encoded characters to form a word. We investigated encoding of Brisaboa et al. (2003) and found promising alternatives.

Our approach is based on the Tagged Huffman encoding (Silva de Moura et al., 2000), where a pair of '0' digits is the signal and the digit '1' is the tag of beginning and end code, the only difference is that for our approach, we need a shorter version to reduce the size of input matrix, so we choose to use only one digit 0 instead of two for each char, marking the beginning and the end of each char code with a digit 1, the same way.

As in the approach by Silva de Moura et al. (2000), the coding we employ has the following advantages (Brisaboa et al., 2003):

1. No character code is a prefix of another, that is, the match is one-to-one.

2. It allows a direct search, that is, to search for a word in an encoded document, just encode the word and use traditional methods of comparing strings with the encoded document.

3. This encoding approach is a compression technique, so it also allows saving already encoded text documents permanently using a binary system, requiring less storage space.

These advantages become attractive especially if the goal is to extract knowledge about files in a repository to perform various classifications on different dimensions with the same files.

A possible advantage of this encoding over others strategies that use a word as an atomic representation of text is better respond to unseen words on training data, once that the network at least have some prefix to guess the meaning of the word, the same way we humans do. This is especially interesting in languages where there are a lot of declensions like Portuguese, Spanish, Italian, French, Russian, German, Arabic and Turkish, for example.

The coding procedure is not restricted to any size of vocabulary, the only problem is that less frequent characters will generate bigger codes, consequently, bigger encoding matrix. If your database has a lot of them, you could use a higher word code size.

Our main contribution is demonstrate that such approach reduces the dimensionality of the encoded matrix, substantially reducing training time and allow the use of devices with lower computational power, remaining with competitive accuracy.

### 3.2 Encoding Procedure

In all of our experiments, we used the following procedure to encode words:

- *Obtaining a character frequency rank*: We read the text database and count the frequency of each character, generating a list sorted by frequency of occurrence. Then we create a rank with only the relative positions of the characters. For a given language, this rank is quite stable since only the order of the rank is used. This means that if all documents are in the same idiom, this procedure can be replaced by a list with the characters rank of frequency for that language.

- *Creating a mapping from character to compress code*: To encode each character, we insert the digit 0 in the same amount of the rank position of the character, between a 1 digit to signal the begin and another 1 digit to signal the end of the code. Table 1 have some examples of encoded characters.

To encode each word, we just concatenate the codes of each character. As an example, we provide in Table 2 some examples of plain text words and their corresponding encoding.

Given a document, we consider that it is composed of words, being those any set of characters limited by the space character. This means 'word' could be math equations, web addresses, LaTeX code, computer programming commands, etc. In Table 2 we can see that this encoding could represent words with the same prefix in vectors that have same initial coordinates. Slangs like *tl;dr : too long, did not read* and *u2 : you too* as well mathematical expressions like $e^a$ could be manipulated.

Table 1: Example of coding using English language ranking of characters. Characters shown are the ones used in the subsequent examples.

| CHARACTER | FREQ. RANK | COMPRESSED ENCODING |
|:---:|:---:|:---|
| ␣ | 0 | 11 |
| e | 1 | 101 |
| a | 2 | 1001 |
| t | 3 | 10001 |
| i | 4 | 100001 |
| s | 5 | 1000001 |
| n | 7 | 100000001 |
| r | 8 | 1000000001 |
| d | 10 | 100000000001 |
| h | 11 | 1000000000001 |
| c | 12 | 10000000000001 |
| g | 17 | 1000000000000000001 |
| $\vdots$ | $\vdots$ | $\vdots$ |
| | $n$ | $'1' + n \times '0' + '1'$ |

Table 2: Example of coding using English language ranking of characters. Prefix common to more than an word are underscored

| TEXT | ENCODED TEXT |
|:---|:---|
| science | 1000001100000000000001100001101100000000110000000000001101 |
| scientist | 1000001100000000000001100001101100000000110001100001100000110001 |
| art | 1001100000000110001 |
| artist | 10011000000001100011000011000000110001 |
| tl;dr | 10001100000000011000000000000000000000000000011000000000011000000001 |
| u2 | 10000000000000011000000000000000000000000000000001 |
| $e^a$ | 101100000000000000000000000000000000000000000000000000000000011001 |

In a matrix of $number\ of\ words \times code\ size$ representing the document, each row represents a properly encoded word, where the code is embedded with its first symbol in the first column. Columns that were not occupied are padded with 0, larger codes are represented up to the chosen limit. Unoccupied lines are filled with 0 and larger documents are represented only up to the chosen maximum number of words, ignoring the last remaining ones. As an example, we represent a document in an $8 \times 65$ matrix in Figure 1.

In the example in Figure 1 we used a $8 \times 65$ matrix (520 elements) to encode the text with a certain amount of slack. At the very least, we would need $7 \times 64$ (448 elements). In contrast, the approach of Zhang et al. (2015) would employ at least $32 \times 69$ (2208) elements to represent the same sentence.

In our experiments, 256 coordinates were enough to represent 99.5% of the words from one of the databases studied. In all datasets studied in this paper, we choose 128 as a limit of words to represent a document, encoding each text in a $128 \times 256$ matrix.

### 3.2.1 CONVOLUTIONAL NETWORK MODEL

As mentioned before, this work was prompted by the results of Zhang et al. (2015). In their original approach, they encode each character using one-hot encoding in a vocabulary of 69 elements. The non-space characters are letters, numbers and punctuation. The model is composed of 9 layers, 6 of convolutions and 3 fully connected. Their architecture is described in Table 3.

They used stochastic gradient descent (SGD) with a minibatch of size 128, using momentum 0.9 and initial step size 0.01 which is halved every 3 epochs for 10 times. So, their results were obtained in at least 30 epochs.

10000000011011000001101100110000000011000000000000110000000000010
1000011000001000000000000000000000000000000000000000000000000000
1001100000001000000000000000000000000000000000000000000000000000
1001100000000110001000000000000000000000000000000000000000000000
1001100000000110000000000010000000000000000000000000000000000000
1001000000000000000000000000000000000000000000000000000000000000
1000001100000000000011000011011000000011000000000000001101000000000
0000000000000000000000000000000000000000000000000000000000000000

Figure 1: Matrix encoding of sentence 'Research is an art and a science'. The text is encoded one word per row. Each word is underscored for easy identification.

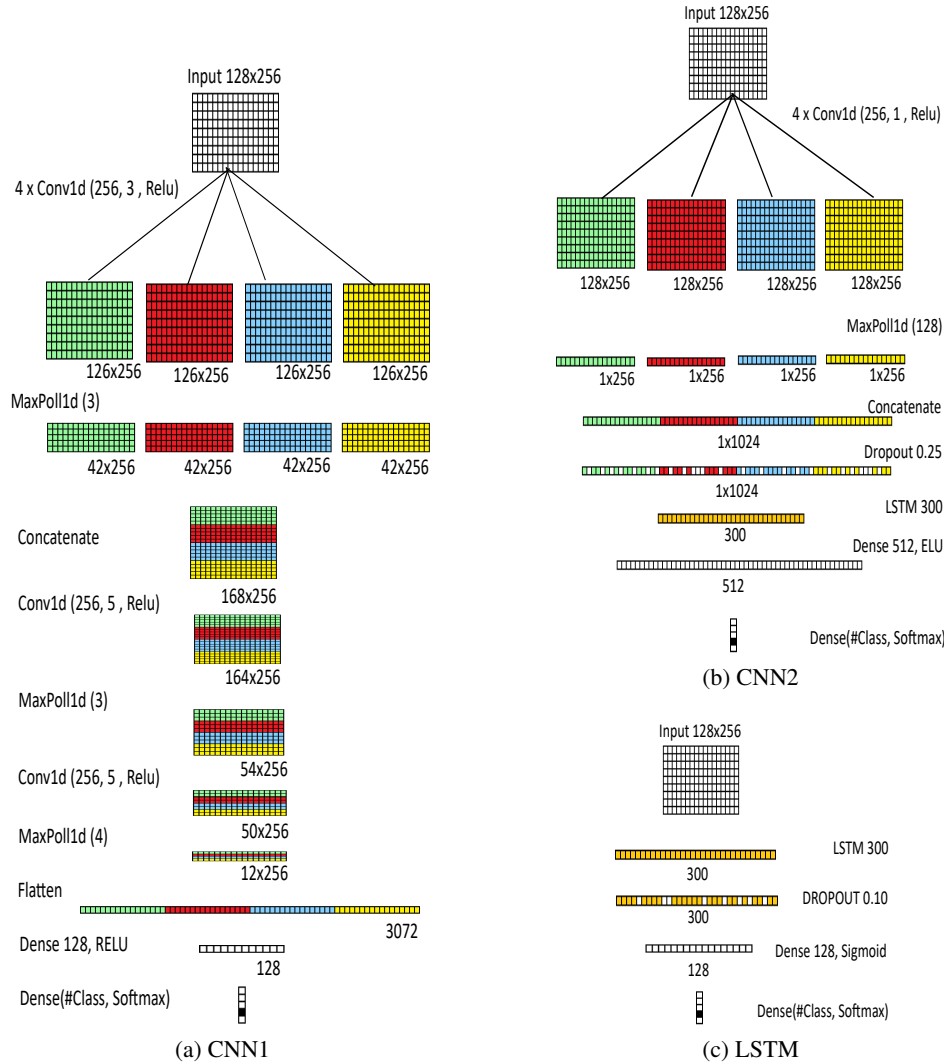

Figure 2: Network architectures used coupled with the compressed encoding.

### 3.3 NEURAL NETWORK ARCHITECTURES

To verify the efficiency of the encoding procedure, we realized 3 experiments, named CNN1 , CNN2 and LSTM:

Table 3: Architectures of the 'large' and 'small' CNNs used by Zhang et al. (2015).

**CONVOLUTIONS**

| Layer | Large Feature | Small Feature | Kernel | Poll |
|-------|---------------|---------------|--------|------|
| 1 | 1024 | 256 | 7 | 3 |
| 2 | 1024 | 256 | 7 | 3 |
| 3 | 1024 | 256 | 3 | — |
| 4 | 1024 | 256 | 3 | — |
| 5 | 1024 | 256 | 3 | — |
| 6 | 1024 | 256 | 3 | 3 |

**FULLY CONNECTED**

| Layer | Large Feature Out | Small Feature Out | Dropout |
|-------|-------------------|-------------------|---------|
| 7 | 2048 | 1014 | .5 |
| 8 | 2048 | 1014 | .5 |
| 9 | Depends on the problem | | |

### 3.4 CNN1 TOPOLOGY

At first, we choose a network architecture that we usually use to classify text using an embedding created by word2vec (Mikolov et al., 2013), the only difference is that instead of 300 features, we reduce the input size to 256. This architecture we named CNN1. It is based on concatenation of convolutions in a shallow way, inspired by work of Kim (2014), who achieve state of the art results for some databases.

We trained this model for 5 epochs. The neural network architecture CNN1 is summarized in Figure 2a as a diagram.

### 3.5 CNN2 TOPOLOGY

Prompted by the positive outcome of CNN1, we decided to investigate others possible architectures. We created another shallow but wide convolution architecture following the recommendations of Zhang & Wallace (2015) for choosing parameters, executing training on dataset ag_news, for being the smallest of our test datasets. This architecture is composed by:

- *Convolution width filter*: a combination of region sizes near the optimal single best region size outperforms using multiple region sizes far from the optimal single region size (Zhang & Wallace, 2015). We scan the width from 1 to 7 comparing accuracy performance. In these evaluations, the convolution of width 1 was a better option.

- *Pooling size*: max pooling consistently performs better than alternative strategies for the task of sentence classification (Zhang & Wallace, 2015).

The neural network architecture CNN2 is summarized in Figure 2b as a diagram

### 3.6 LSTM TOPOLOGY

To illustrate the possibilities of applying the proposed encoding, we did a experiment using LSTMs (Hochreiter & Schmidhuber, 1997), similar to LSTM model described by Zhang & LeCun (2015), the difference is that instead of using word2vec embedding (Mikolov et al., 2013), we used our representation. The architecture is very simple: an input layer of $126 \times 256$, followed by a LSTM layer of 300, a Dropout layer of .10 (Srivastava et al., 2014), a fully connected layer of 128 units and a softmax layer.

We trained this model for 5 epochs. This architecture is in general, twice slower than CNN1. The neural network architecture LSTM is summarized in Figure 2c as a diagram.

## 4 EXPERIMENTAL STUDY

An essential part of this work is to contrast our proposal both within the context of character-level text classification and with other state-of-the-art approaches.

The databases used were the same as those cited in an article by Zhang et al. (2015), where there is an extensive description of them.[1] A detailed analysis of these datasets is out of the scope of this paper, instead, we will only summarize the main characteristics:

- **AG's news**: categorized news articles from more than 2000 news sources. Four classes (World, Sports, Business, SciTech).The dataset contains 120k train samples and 7.6k test samples equally distributed (Zhang et al., 2015).

- **Sogou news**: categorized news articles originally in Chinese. Zhang et al. (2015) applied the *pypinyin* package combined with *jieba* Chinese segmentation system to produce Pinyin – a phonetic romanization of Chinese. Five classes (sports, finance, entertainment, automobile and technology). The dataset contains 450k train samples and 60k test samples equally distributed (Zhang et al., 2015).

- **DBpedia**: title and abstract from Wikipedia articles available in DBpedia crowd-sourced community (Lehmann et al., 2015). Fourteen non-overlapping classes from DBpedia 2014. The dataset contains 560k train samples and 70k test samples equally distributed (Zhang et al., 2015).

- **Yelp full**: sentiment analysis from the Yelp Dataset Challenge in 2015[2]. Five classes representing the number of stars a user has given.The dataset contains 560k train samples and 38k test samples equally distributed. (Zhang et al., 2015).

- **Yelp polarity**: sentiment analysis from the Yelp Dataset Challenge in 2015[2]. The original data is transformed into a polarity problem. Rating of 1 and 2 stars are represented as Bad and 4 and 5 as Good. The dataset contains 560k train samples and 50k test samples equally distributed. (Zhang et al., 2015).

- **Yahoo! answers**: questions and their answers from Yahoo! Answers. Ten classes (Society & Culture, Science & Mathematics, Health, Education & Reference, Computers & Internet, Sports, Business & Finance, Entertainment & Music, Family & Relationships, Politics & Government). Each sample contains question title, question content and best answer.The dataset contains 1,400k train samples and 60k test samples (Zhang et al., 2015).

- **Amazon full**: sentiment analysis from Amazon reviews dataset from the Stanford Network Analysis Project (SNAP) (McAuley & Leskovec, 2013). Five classes representing the number of stars a user has given. The dataset contains 3,000k train samples and 650k test samples.(Zhang et al., 2015).

- **Amazon polarity**: sentiment analysis from Amazon reviews dataset from the Stanford Network Analysis Project (SNAP) (McAuley & Leskovec, 2013). Two classes, rating of 1 and 2 stars are represented as Bad and 4 and 5 as Good. The dataset contains 3,600k train samples and 400k test samples equally distributed.(Zhang et al., 2015).

The baseline comparison models are the same of Zhang et al. (2015) where there is an extensive description of them, we just reproduce their results, the only difference is that they report loss error and for better comprehension, we translated it to accuracy. In Zhang et al. (2015) there is an extensive description of them. In this paper, we just summarize the main information:

- **Bag of Words (BoW) and its term-frequency inverse-document-frequency (BoW TFIDF)**: For each dataset, they selected 50,000 most frequent words from the training subset. For the normal bag-of-words, they used the counts of each word as the features and for the TFIDF they used the counts as the term-frequency (Zhang et al., 2015).

---

[1]For the sake of replicability we have made all the datasets available via https://drive.google.com/open?id=1o5CNT0UHuFfHBxC-Mz4ImFpN2-Lcllmx.

[2]https://www.yelp.com/dataset/challenge

Table 4: Training environment and parameters

| DESCRIPTION | PARAMETERS | OBSERVATIONS |
|---|---|---|
| Neural Net Lib. | Keras 2.0 | |
| Tensor Backend | Theano 0.9 | |
| GPU Interface | Cuda 8 | with cuBLAS Patch Update |
| CNN optimizer | Nvidia Cudnn 5.1 | |
| Program. Lang. | Python 3.6 | using Anaconda 4.4.0 |
| Superbatch | 10000 | Number of matrixes sent to gpu each time |
| Minibatch | 32 | Batch to update the network weights |
| Optimizer | ADAM (Zeiler, 2012) | $lr = 10^{-3}$, $\beta_1 = 0.9$, $\beta_2 = 0.999$, $\epsilon = 10^{-8}$ |
| Epochs | 5 , 12 | |
| Op. System | Windows 10 | |
| GPU | Nvidia GeForce 1080ti | |
| RAM Memory | 16 GB | |

- **Bag-of-ngrams (Ngrams) and its TFIDF (Ngrams TFIDF)**: The bag-of-ngrams models were constructed by selecting the 500,000 most frequent n-grams (up to 5-grams) from the training subset for each dataset. The feature values were computed the same way as in the bag-of-words model (Zhang et al., 2015).

- **Bag-of-means on word embedding**: an experimental model that uses $k$-means on word2vec (Mikolov et al., 2013) learned from the training subset of each dataset, and then used these learned means as representatives of the clustered words. Took into consideration all the words that appeared more than 5 times in the training subset. The dimension of the embedding is 300. The bag-of-means features are computed the same way as in the bag-of-words model. The number of means is 5000 (Zhang et al., 2015).

- **Long Short Term Memory (LSTM)**: The LSTM (Hochreiter & Schmidhuber, 1997) model used is word-based, using pretrained word2vec (Mikolov et al., 2013) embedding of size 300. The model is formed by taking a mean of the outputs of all LSTM cells to form a feature vector, and then using multinominal logistic regression on this feature vector. The output dimension is 512 (Zhang et al., 2015).

For all the experiments, we used the environment and parameters settings listed in Table 4. Besides the encoding procedure, we do not use any preprocessing strategy except the use of lowercase letters. No data enhancement technique was employed.

All the results and comparison with traditional models and the approaches of Zhang et al. (2015) is shown in Table 5.

As tabular data are a hard to grasp, we have decided to go for a graphical representation of the results. In particular, from the previous results, we have selected the large and small architectures of Zhang et al. (2015) and our CNN1, CNN2 and LSTM. Those values of accuracy were then scaled to the interval $[0, 1]$ for every dataset, being 0 the worst and 1 the best performance among all models, including tradicional models . The outcome of this process is represented on Figure 3. On Table 6, we did a running time comparison, based on reports available by Zhang et al. (2015).

The main objective of this research was to evaluate the possibility of using a coding approach that contemplated the construction of words using characters as basic tokens. Our main contribution is demonstrate that such approach allows reducing the dimensionality of the encoding matrix, thus allowing substantially shorter optimization times and the use of devices with lower computational power. Some datasets of text have peculiarities that were not addressed by word frequency methods (i.e. BoW and word2vec), like declensions and new vocabulary. The article of Zhang et al. (2015) was a great innovation in this regard. However, the training times are still obstacles to the most effective use of the technique. We thought of a better representation should be a solution.

To make a comparison, perform the training of architecture CNN1 with an output of 4 classes make necessary to optimize 1,837,188 parameters. As a comparison, in the architecture suggested by Zhang et al. (2015) it is necessary to optimize 11,337,988 parameters (Zhang et al., 2015).

Table 5: Test set accuracy comparison among traditional models the Zhang et al. (2015) models 'large' and 'small' and our approaches when applied to the AG's news (AG), Sogou news (SOGOU), DBpedia (DBP), Yelp polarity (YLP-P), Yelp full (YLP), Yahoo! answers (YAH), Amazon full (AMZ) and Amazon polarity (AMZ-P) datasets.

| MODEL | | AG | SOGOU | DBP | YLP-P | YLP | YAH | AMZ | AMZ-P |
|---|---|---|---|---|---|---|---|---|---|
| BoW | | 88.81 | 92.85 | 96.61 | 92.24 | 57.99 | 68.89 | 54.64 | 90.40 |
| BoW TFIDF | | 89.64 | 93.45 | 97.37 | 93.66 | 59.86 | 71.04 | 55.26 | 91.00 |
| Ngrams | | 92.04 | 97.08 | 98.63 | 95.64 | 56.26 | 68.47 | 54.27 | 92.02 |
| Ngrams TFIDF | | 92.36 | 97.19 | 98.69 | 95.44 | 54.80 | 68.51 | 52.44 | 91.54 |
| Bag-of-Means | | 83.09 | 89.21 | 90.45 | 87.33 | 52.54 | 60.55 | 44.13 | 81.61 |
| LSTM | | 86.06 | 95.18 | 98.55 | 94.74 | 58.17 | 70.84 | 59.43 | 93.90 |
| Zhang et al. | Large | 87.18 | 95.12 | 98.27 | 94.11 | 60.38 | 70.45 | 58.69 | 94.49 |
| | Small | 84.35 | 91.35 | 98.02 | 93.47 | 59.16 | 70.16 | 59.47 | 94.50 |
| Ours | CNN1 | 87.67 | 95.16 | 97.93 | 92.04 | 58.00 | 68.10 | 58.09 | 93.69 |
| | CNN2 | 91.43 | 93.96 | 98.03 | 91.53 | 57.03 | 70.24 | 55.72 | 91.23 |
| | LSTM | 88.38 | 94.52 | 98.34 | 93.18 | 59.71 | 70.27 | 59.79 | 94.35 |

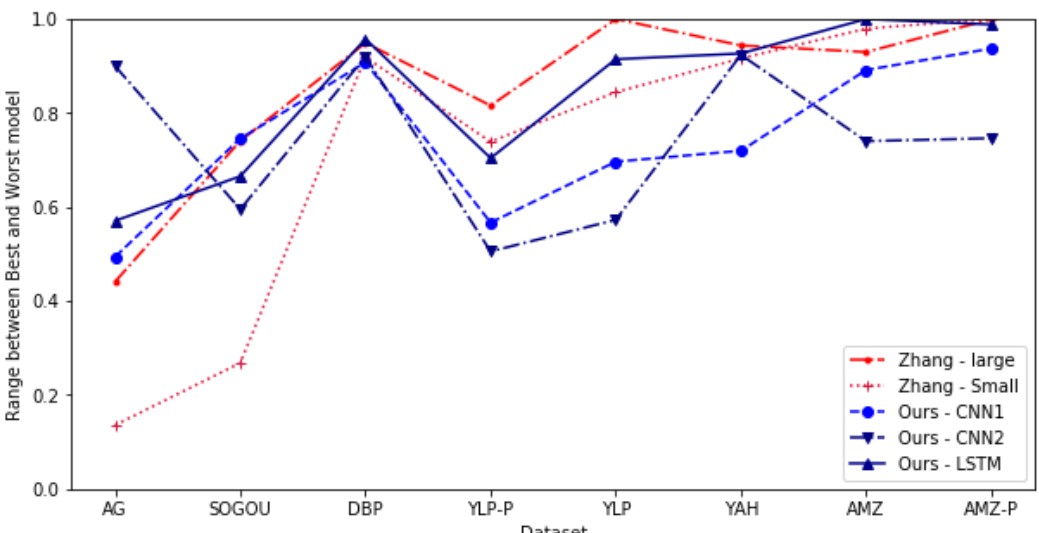

Figure 3: Performance re-scaled in the range between best and worst model comparing (Zhang et al., 2015) and our approaches when applied to the AG's news (AG), Sogou news (SOGOU), DBpedia (DBP), Yelp polarity (YLP-P), Yelp full (YLP), Yahoo! answers (YAH), Amazon full (AMZ) and Amazon polarity (AMZ-P) datasets.

As one of our concerns was to make our proposal as usable as possible on commodity hardware, we focused our studies in that hardware configuration. The major bottleneck for analyzing this large amount of matrix-encoded text is the need for intensive use of RAM. Our approach generates a $128 \times 256$ matrix, smaller than the $1014 \times 69$ generated by Zhang et al. (2015). In spite of that, a large set of them quickly occupies the available RAM on a 'regular' personal computer. On the machine we used, there were 16 GB available, which is not uncommon in modern personal computers. Therefore, the use of generators to control the number of matrices generated and sent to GPU is an important detail in the implementation of this optimization algorithm. If your computer has only 8 GB of RAM or less, it will be necessary to reduce the number of superbatchs to fit the memory.

The results obtained strongly indicate that the use of this coding is a possibility. We emphasize that we used is a fairly simple network, enough to demonstrate the feasibility of the encoding approach with the time and computational resources that we have.

Table 6: Time per epoch as reported by Zhang et al. (2015) models and the ones used by CNN1 and CNN2 on an NVidia GeForce 1080ti GPU.

| DATASET | TRAIN/TEST SIZE | ZHANG ET AL., 2015 | | | OURS | |
|---|---|---|---|---|---|---|
| | | LARGE | SMALL | CNN1 | CNN2 | LSTM |
| AG's news | 120 k / 7.6 k | 1 h | 3 h | 3 min | 4 min | 7 min |
| Sogou news | 450 k / 60 k | N/A | N/A | 23 min | 27 min | 42 min |
| DBPedia | 560 k / 70 k | 2 h | 5 h | 18 min | 20 min | 36 min |
| Yelp polarity | 560 k / 38 k | N/A | N/A | 21 min | 31 min | 47 min |
| Yelp full | 650k / 50 k | N/A | N/A | 27 min | 46 min | 48 min |
| Yahoo! answers | 1,400 k / 60 k | 8 h | 1 days | 47 min | 55 min | 1 h 37 min |
| Amazon full | 3,000 k / 650 k | 2 days | 5 days | 2 h | 2 h 30 min | 3 h 53 min |
| Amazon polarity | 3,600 k / 400 k | 2 days | 5 days | 2h 13 min | 2 h 31 min | 4 h 18 min |

The results are very competitive with the approach o Zhang et al. (2015) and traditional techniques. We see even that we could find parameters that achieve excellent performance on AG's news dataset, following the suggestions of Zhang & Wallace (2015). One of advantage of a faster algorithm is that if your dataset is not so big, you could scan the feature width to find a solution that optimizes accuracy. Another advantage is the possibility to realize k-folds validation, to have a better perspective on how well it will perform for your specific dataset on real life.

Another interesting point is that using a LSTM layer with the proposed encoding, we achieved similar, sometimes better results than using an embedding using word2vec (Mikolov et al., 2013) as proposed by Zhang et al. (2015). Our approach by its own nature take into account morphological aspects of text while word2vec uses pre-trained vectors representing co-ocurrency of words on a big corpus of text. Being able to take into account character information even in recurrent networks, we show that this representation is not limited to the domain of CNN or neural networks, for that matter.

Although our LSTM architecture is twice slower than our CNN1 and CNN2 topologies, it consistently outperform them. This indicates that temporal dependence among of words are important, so, potentially other architectures can generate better results taking this information to account and this is a direction that should be explored. In addition to that, the dimensionality reduction achieved by our encoding enables several other architectures and methods to be verified in a reasonable timeframe.

We are certain that our algorithm implementation could be even faster. For instance, when using a GPU Geforce 1080ti and a CNN1 architecture, each of the superbatch of 10,000 arrays have its weights updated in 30 seconds. Only 6 seconds is consumed by GPU, another 24 seconds is spent in encoding all the matrix and delivery it on the GPU. Using a multithread strategy could help in this regard.

## 5 FINAL REMARKS

In this paper, we have proposed an efficient character-level encoding for text derived from the Tagged Huffman (Silva de Moura et al., 2000) information compression method and applied it as input preprocessing step for character-level CNN text classification.

We have shown that using this compression technique is a convenient possibility to represent text in a convolutional deep learning setup for text classification. This is particularly important as encoding text using characters can be relevant when dealing with less curated texts datasets, as it is robust to spelling errors, typos, slangs, language variations, and others usual characteristics of Internet texts.

This novel text encoding allow to represent larger portions of texts in a compact form while preserving the ability to encode any word, even those not present in the training dataset ones. Furthermore, for being compact yet sparse, this approach dramatically reduces the time required for training CNNs and, therefore, makes the application of this novel approach more accessible. This opens the door to more complex applications, the use of devices with lower computational power and the exploration of other approaches that can be coupled with this input representation.

The experimental studies carried out coupled the encoding with two convolutional neural networks architectures and a recurrent LSTM architecture. These experiments showed that we managed to achieve a performance similar to the one achieved by Zhang et al. (2015) in a fraction of the training time even if we employed a simpler hardware setup. Furthermore, with our results, we endorse Zhang et al. (2015) conclusions, which state that language can be treated as a signal, no different from any other.

It should be noted that the main objective of the paper was to show the viability of the text encoding, not producing better results *per se*. Consequently, we have focused our efforts on the comparative study. Probably, custom neural network architectures should be devised to this new encoding with that purpose. Our results indicate that combining it with LSTMs is a promising direction in order to overcome the fixed matrix size limitation. In the near future, we will focus on devising these new architectures that may further improve the results.

This study also opens a door to other information-theoretic-based methods for information representation to be used to create a numerical representation of texts. It must be outlined the fact that this compact numeric representation of text is not limited to the domain of CNN or neural networks, for that matter. It could be interesting to assess its impact as a preprocessing step, perhaps with a minor modification, for other classification algorithms.

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
