# OpenReview forum: "Compact Encoding of Words for Efficient Character-level Convolutional Neural Networks Text Classification"
_ICLR.cc/2018/Conference — Reject_

### Official Review · AnonReviewer3 · 2017-11-11
**Main idea lacks significance**

**Rating:** 4
**Confidence:** 5

**Review:**

The paper proposed to encode text into a binary matrix by using a compressing code for each word in each matrix row. The idea is interesting, and overall introduction is clear.

However, the work lacks justification for this particular way of encoding, and no comparison for any other encoding mechanism is provided except for the one-hot encoding used in Zhang & LeCun 2015. The results using this particular encoding are not better than any previous work.

The network architecture seems to be arbitrary and unusual. It was designed with 4 convolutional layers stacked together for the first layer, while a common choice is to just make it one convolutional layer with 4 times the output channels. The depth of the network is only 5, even with many layers listed in table 5.

It uses 1-D convolution across the word dimension (inferred from the feature size in table 5), which means the convolutional layers learn intra-word features for the entire text but not any character-level features. This does not seem to be reasonable.

Overall, the lack of comparisons and the questionable choices for the networks render this work lacking significance to be published in ICLR 2018.

---

> ### Author Response · Authors · 2017-12-28
> **We did a updated version addressing your key recommendations**
>
> Thank you for your comments.
>
>     We created an updated version where we did our best to improve the quality of our presentation and express it in a clear way.
>
>     In this new version, we provide a better description and justification of our proposal. In particular, we discuss why we took some decisions regarding the encoding. Similarly, we now include an additional neural network architecture in the comparative experiments. For completeness, we included a more detailed description of datasets and base models involved in the experiments.
>
>     Regarding encodings comparison, in the initial stage of our research, we investigated others encodings, mainly Huffman and End Tag Dense Codes- ETDC. We discarded these because we did not found a way to represent words in a distinct way once we concatenate each char code. We made more clear this step in the manuscript and still working in a way to modify ETDC to make it distinct for each word.
>
>     Our main concern with this paper is show that a more compact encoding procedure could reduce the computational footprint of using words codified character-by-character in text classification.  Character-based text classification allows to handle less curated texts or texts in languages that have a lot of declensions and with poor or none a priori pre-trained language modules (ala word2vec).
>
>     In terms of classification performance, we have matched the results of Zhang & LeCun (2015) which represent the state of the art in this area and the departing point of this work. It should be noted that like them, we also beat traditional methods. In this regard, in order to improve the readability of the paper, we changed the comparison metric to accuracy, different of Zhang & LeCun (2015) that report error loss ($(1-accuracy)\times100$).
>
>     On the other hand, in terms of computational footprint, our approach is much faster than Zhang & LeCun (2015). We find that is is a relevant result as it makes the approach suitable for extended use. We are providing along with our paper the supporting code. We expect that with the collaboration of the community a streamlined implementation can be obtained and even better times will be reached.
>
>     To our knowledge, our approach is the first who try to codify words into vectors using only characters composition in a sparse way. We are aware that there is room for improvement. In the process of updating the paper, we included another neural network with positive results. Nevertheless, this direction should be properly explored in the future. In this paper, we focused mainly on the comparison with the previous results by Zhang & Lecun (2015) but we are confident that other architectures will yield better results.
>
>     Our main line of research is educational data mining where it is crucial to be able to handle texts produced by students, with equations, orthographic errors and not-so-formal language. In this scenario, we have a lot of interest in building better and faster solutions build upon the character-based approach originally put forward by Zhang & Lecun (2015).
>
>     We appreciate that you take a moment an revise again that paper under the light of these comments and modifications.

---

### Official Review · AnonReviewer2 · 2017-11-27
**The manuscript needs updating. The current state is not good enough.**

**Rating:** 3
**Confidence:** 5

**Review:**

The manuscript proposed to use prefix codes to compress the input to a neural network for text classification. It builds upon the work by Zhang & LeCun (2015) where the same tasks are used.


There are several issues with the paper and I cannot recommend acceptance of the paper in the current state.
- It looks like it is not finished.
- the datasets are not described properly.
- It is not clear to me where the baseline results come from.
 They do not match up to the Zhang paper (I have tried to find the matching accuracies there).
- It is not clear to me what the baselines actually are or how I can found more info on those.
- the results are not remarkable.

Because of this, the paper needs to be updated and cleaned up before it can be properly reviewed.

On top of this, I do not enjoy the style the paper is written in, the language is convoluted.
For example: “The effort to use Neural Convolution Networks for text classification tasks is justified by the possibility of appropriating tools from the recent developments of techniques, libraries and hardware used especially in the image classification “
I do not know which message the paper tries to get across here.
As a reviewer my impression (which is subjective) is that the authors used difficult language to make the manuscript look more impressive.
The acknowledgements should not be included here either.

---

> ### Author Response · Authors · 2017-12-28
> **We did a updated version addressing your key recommendations**
>
>   Thank you very much for your time and constructive comments. We have addressed the issues pointed out in your remarks and tried to make more evident the contributions of our work. We made important improvements on the quality of the text, the description of our proposal and presentation of the experimental results.
>
>     In this new version, we provide a better description and justification of our proposal. In particular, we discuss why we took some decisions regarding the encoding. Similarly, we now include an additional neural network architecture in the comparative experiments. For completeness, we included a more detailed description of datasets and base models involved in the experiments.
>
>     Our main concern with this paper is show that a more compact encoding procedure could reduce the computational footprint of using words codified character-by-character in text classification.  Character-based text classification allows to handle less curated texts or texts in languages that have a lot of declensions, and with poor or none a priori pre-trained language modules (ala word2vec).
>
>     In terms of classification performance, we have matched the results of Zhang & LeCun (2015) which represent the state of the art in this area and the departing point of this work. It should be noted that like them, we also match traditional methods. In this regard, in order to improve the readability of the paper we changed the comparison metric to accuracy, different of Zhang & LeCun (2015) that report error loss ($(1-accuracy)\times100$).
>
>     On the other hand, in terms of computational footprint, our approach is much faster than Zhang & LeCun (2015). We find that is is a relevant result as it makes the approach suitable for extended use. We are providing along with our paper the supporting code. We expect that with the collaboration of the community a streamlined implementation can be obtained and even better times will be reached.
>
>     To our knowledge, our approach is the first who try to codify words into vectors using only characters composition in a sparse way. We are aware that there is room for improvement. In the process of updating the paper we included another neural network with positive results. Nevertheless, this direction should be properly explored in the future. In this paper we focused mainly on the comparison with the previous results by Zhang & Lecun (2015) but we are confident that other architectures will yield better results.
>
>     Our main line of research is educational data mining where it is crucial to be able to handle texts produced by students, with equations, orthographic errors and not-so-formal language. In this scenario, we have a lot of interest in building better and faster solutions build upon the character-based approach originally put forward by Zhang & Lecun (2015).
>
>     We appreciate that you take a moment an revise again that paper under the light of these comments and modifications.
>
>     Regarding the writing style, we really apologize. Sometimes is difficult to express yourself in a foreign language. We did your best in this updated version to not give you these impressions.

---

> > ### Comment · AnonReviewer2 · 2018-01-12
> > **Full re-write**
> >
> > After looking at the revision, the manuscript looks in a much better shape at this point.
> > However, due to the amount of  changes,
> > I believe it has to go trough a full review process again as I mentioned in the original review.
> >
> > Therefore I stand by my original opinion the paper cannot be accepted now.
> >
> > If the authors are thinking to re-submit this manuscript, I think they should focus on the following:
> >
> > - Check the literature on the datasets and compare to more recent approaches than Zhang & LeCun such that the baselines are the current state of the art. I am not so familiar with these datasets so I do not know the current best approaches for these datasets.
> >
> > - Polish the language. After quickly reading through the manuscript, I found several more strange formulations.
> >
> > - The times per experiment from Zhang & LeCun should be replaced by current (efficient) re-implementations on the same hardware. Since 2015, we have made advances in hardware and software libraries. Therefore the measurements from 2015 and 2017/2018 are not directly comparable.

---

### Official Review · AnonReviewer1 · 2017-11-28

**Rating:** 2
**Confidence:** 5

**Review:**

This paper proposes a new character encoding scheme for use with character-convolutional language models. This is a poor quality paper, is unclear in the results (what metric is even reported in Table 6), and has little significance (though this may highlight the opportunity to revisit the encoding scheme for characters).

---

> ### Author Response · Authors · 2017-12-28
> **We did a updated version where we did our best to improve the quality of our presentation**
>
> Thank you for your time for helping us to better express our findings.
>
>     We made important improvements in the quality of the text, the description of our proposal and presentation of the experimental results.
>
>     In this new version, we provide a better description and justification of our proposal. In particular, we discuss why we took some decisions regarding the encoding. Similarly, we now include an additional neural network architecture in the comparative experiments. For completeness, we included a more detailed description of datasets and base models involved in the experiments.
>
>     Our main concern with this paper is show that a more compact encoding procedure could reduce the computational footprint of using words codified character-by-character in text classification.  Character-based text classification allows to handle less curated texts or texts in languages that have a lot of declensions and with poor or none a priori pre-trained language modules (ala word2vec).
>
>     In terms of classification performance, we have matched the results of Zhang & LeCun (2015) which represent the state of the art in this area and the departing point of this work. It should be noted that like them, we also beat traditional methods. In this regard, in order to improve the readability of the paper, we changed the comparison metric to accuracy, different of Zhang & LeCun (2015) that report error loss ($(1-accuracy)\times100$).
>
>     On the other hand, in terms of computational footprint, our approach is much faster than Zhang & LeCun (2015). We find that is is a relevant result as it makes the approach suitable for extended use. We are providing along with our paper the supporting code. We expect that with the collaboration of the community a streamlined implementation can be obtained and even better times will be reached.
>
>     To our knowledge, our approach is the first who try to codify words into vectors using only characters composition in a sparse way. We are aware that there is room for improvement. In the process of updating the paper, we included another neural network with positive results. Nevertheless, this direction should be properly explored in the future. In this paper, we focused mainly on the comparison with the previous results by Zhang & Lecun (2015) but we are confident that other architectures will yield better results.
>
>     Our main line of research is educational data mining where it is crucial to be able to handle texts produced by students, with equations, orthographic errors and not-so-formal language. In this scenario, we have a lot of interest in building better and faster solutions build upon the character-based approach originally put forward by Zhang & Lecun (2015).
>
>     We appreciate that you take a moment an revise again that paper under the light of these comments and modifications.

---

### Public Comment · (anonymous) · 2017-12-03
**question table 6**

Hello,

I am part of a team at McGill University participating in the ICLR 2018 reproducibility challenge.
We are currently trying to reproduce results from your paper.

We have a few questions and would be very thankful if you could answer them:
1) what is the loss function used to compare models on table 6 ?
2) can you provide more details about your implementation of the traditional models described in table 6?

Thank you in advance.

---

> ### Author Response · Authors · 2017-12-03
> **Table 6**
>
> Hello
>
> Thank you for you interest in this approach.
>
> Table 6 results are testing error, or 1- accuracy*100.
>
> Theses results came from Table 4 in Zhang, Zhao, LeCun paper: https://arxiv.org/pdf/1509.01626.pdf .  We just compare their results with ours.

---

### Public Comment · (anonymous) · 2017-12-12
**Activation Function**

Hello again,

I am part of the same McGill team working on your paper. Thank you very much for taking the time to answer our question last time.

We have another question:
What activation functions did you use in the CNN ?

Thank you in advance.

---

> ### Author Response · Authors · 2018-01-03
> **we used RELU**
>
> we have not seen your question in time to help you on the reproducibility challenge, but you guess right, we used RELU
>
> We are glad that you could replicate the main findings using just our instructions, but in this new version, we tried to give more information on how and why we took some decisions, in a way that could be easier to reproduce the same findings.
>
>  We have prepared a Jupyter/IPython notebook that we publish online along with the paper. We did not publish it yet to not infringe the double-blind review policy.
>
> Thank you

---

### Public Comment · (anonymous) · 2017-12-15
**ICLR2018 Reproducibilty Challenge - Review**

This is an executive summary of a more detailed report that can be found here:
https://de.scribd.com/document/367280305/COMP-551-Project-4-Reproducible-Machine-Learning
This project is part of the ICLR2018 Reproducibility Challenge. The goal of the challenge is to improve the quality of submissions by highlighting the importance of reproducibility. In this review we present the main challenges we faced while reproducing the ideas presented in this paper as well as highlighting which aspects were easily reproducible.

We gathered three of the analyzed data sets, implemented the encoding scheme and the convolutional neural network according to the description given in the paper.

Our results (test error as well as running times) are very similar to the results of the original paper, and we were able to reproduce it for the most part. The test errors achieved on the datasets were the following (original results in parenthesis):
ag_news 11.89 (12.33), dbpedia 2.33 (2.07) & yelp_polarity 7.84 (7.96).
Our training time per epoch (trained on a Nvidia Tesla K80 whereas the authors used a Nvidia 1080ti):
ag_news 4.87 min (3 min), dbpedia 23.11 min (18 min) & yelp_polarity 20.17 min (21 min).

However, as can be seen from these values, we could not reproduce exact results. This was mainly due to the fact, that not all of the details of preprocessing, network architecture and hyperparameters were available to us. In the following, we address the most relevant points we faced in our replication:

The original paper does not contain any direct links or any other information on the data sets used. All the information on the data sets was gathered from papers cited. Therefore, extra work and time had to be spent tracking these data sets down. Once we had found the data sets, we were able to exactly replicate the split into training and test set, because this split was already provided.

The description of the encoding function is very clear and examples make it easy to understand. We're confident in our replication of the encoding process. However, the paper doesn't go into detail on the preprocessing of the data sets. We were unsure how newline characters in the documents were preprocessed for instance. For replication purposes, a detailed description of the preprocessing employed would have been helpful.

The network architecture is presented in form of a table. Some important implementation details are missing (activation functions, loss function used), and others have to be deduced by observing the output dimensions of the individual layers of the network. This makes it difficult to exactly replicate the network the authors used. Again, a more detailed description would have been helpful.

The computing infrastructure (including library versions) used was clearly explained, and even though we did not possess the exact same environment, we believe that one would be able to set up the exact same infrastructure with the information provided.

We did not have access to the code of the authors and therefore had to implement the full model on our own. In some cases we were missing information on parameters or how exactly things were implemented (see above). Our implementation could therefore be different from the one of the original authors, affecting computation times. However, since the paper was pretty clear for the most part, and our results resemble the ones of the authors, we are relatively certain that this has not been a big issue.

Finally, some interaction with the authors was necessary to clarify a few points that were left ambiguous after reading their paper. Some of the results and tables were not described very extensively by the authors and therefore needed clarification. We contacted them using this platform and received a quick answer. We also contacted them about the activation functions used in the network but received no reply until the submission deadline.

This review was created by Seara Chen, Benjamin Paul-Dubois-Taine and Lino Toran Jenner, students of McGill University, Montreal.

---

> ### Author Response · Authors · 2017-12-28
> **We are glad that you could replicate the main findings using just our instructions**
>
> Thank you for your interest in this approach
>
>     We did a updated version addressing your key recommendations.
>
>     We are glad that you could replicate the main findings using just our instructions, but in this new version, We tried to give more information on how and why we took some decisions, in a way that could be easier to reproduce the same findings.
>
>     We have prepared a Jupyter/IPython notebook that we publish online along with the paper. We did not published yet to not infringe the double-blind review policy.
>
>     We invite you to read it again. We are open to any suggestion to better presents these findings.
>
>     Thank you.

---

### Public Comment · (anonymous) · 2017-12-16
**ICLR2018 Reproducibilty Challenge**

Our reproducibility experiment was carried out in the context of the ICLR 2018 Reproducibility Challenge, where various groups are encouraged to reproduce the findings papers submitted to the ICLR 2018 conference. The intended outcome of the initiative is to emphasize the need for reproducibility in the fast-growing field of machine learning.

As contenders in the reproducibility challenge, we chose this paper, which describes a simple scheme for encoding text-data into matrix form, which we dubbed the CME. The encoding is applied to 8 datasets, which are then used as input to a convolutional neural network with the assumption that the algorithm will only deal with short text excerpts in the classification setting. The main claim of the paper is that similar performance to existing methods can be achieved using the CME, albeit at a fraction of the runtime.

We attempted to replicate these findings on 3 of the 8 datasets. Although the datasets were provided in a clear format and did not require further preprocessing, the contents of the datasets were not described nor were the questions of whether and how the particularities of each dataset affects the performance of the proposed algorithm. First, we implemented the encoding procedure using the specifications described in the paper, and encoded the datasets using the CME scheme. Then, we trained a convolutional neural network using the same architecture. Because neither the datasets nor the code were not supplied along with the conference paper, we attempted to implement both these methods ourselves, relying strictly on the specifications provided in the paper.

It was relatively straightforward for us to use the specifications to reproduce most of the methods described in the work. The encoding procedure is presented punctiliously, and with sufficient examples, that a person with a modest level of expertise in ML could likely implement the encoder from scratch. The authors also provide an in-depth summary of the specifications (i.e. parameters, hyperparameters, computational infrastructure), which greatly facilitates any attempt to reproduce the findings.
However, we started to question our implementation in light of the rather poor performance that we obtained. We were unable to replicate the findings of the paper, achieving accuracies matching those yielded by random multiclass classifiers instead. We reasoned that this poor performance was likely due to our making wrong assumptions about how to apply the encoding scheme to the training and test datasets. Because no substantial information regarding exactly how the encoding scheme was to be applied to the training and test sets, we encoded these datasets in a completely independent manner, which we suspect was not what the authors had intended. As such, we make no claim of inaccurate findings, only that we were not able to properly replicate them given the specifications provided in the paper, the time constraints for the reproducibility experiment, and the lack of publicly available code.

The main strength of the paper lies in 1) the novel CME technique which lends itself well to being used by convolutional neural networks and 2) the detailed specification of both the CME and of the neural network architecture. The language used throughout the report is readable and very helpful in aiding understanding of the methods.
The major drawback of the paper lies in the lack of information regarding the baselines and how exactly the encoding scheme is to be applied on the training and test sets. No adequate baseline methods were mentioned in the paper against which we could compare the promising CME. Furthermore, the nature of the metrics used to evaluate the neural nets were not explicitly stated, as only one number was reported per dataset. Only through careful investigation was it possible for us to determine that these numbers in fact correspond to test accuracies. In addition, it would have been interesting to use a host of metrics, and not just one measure of fitness, to establish performance.

In summary, we would like to commend the authors for their contribution to the field of machine learning. The CME is a promising novel take on the use of convolutional neural nets in the text classification setting. The report provides excellent clarity on the underlying methods. However, since neither the code nor details on how to adequately apply the encoding scheme were supplied, reproducing the major findings of the paper proved to be difficult.

---

> ### Author Response · Authors · 2017-12-28
> **Thank you for your interest in this approach**
>
> Thank you for your interest in this approach
>
>     We did a updated version addressing your key recommendations.
>
>     In this new version, we have tried to give more information on how and why we took some decisions, in a way that could be easier to reproduce the same results.
>
>     We have a Jupyter/IPython notebook ready with all the experiments that we intend to publish online along with the paper. We did not published yet to not infringe the double-blind review policy.
>
>     We kindly invite you to read the paper again. We are open to any suggestion to better presents these findings.
>
>     Thank you.

---

### Decision · Program_Chairs · 2018-01-29
**ICLR 2018 Conference Acceptance Decision**

**Decision:**

Reject

**Comment:**

meta score: 4

The paper has been extensively edited during the review process - the edits are so extensive that I think the paper requires a re-review, which is not possible for ICLR 2018

Pros:
 - potentially interesting and novel approach to prefix encoding for character level CNN text classification
 - some experimental comparisons
Cons:
 - lacks good comparison with the state-of-the-art, which makes it difficult to determine conclusions
 - writing style lacks clarity.

I would recommend that the authors continue to improve the paper and submit it to a later conference.